# Replacing SF$_6$ in Electrical Gas-Insulated Switchgear: Technological Alternatives and Potential Life Cycle Greenhouse Gas Savings in an EU-28 Perspective

**Pieter Billen** [1,*] **, Ben Maes** [2] **, Macarena Larrain** [1,3] **and Johan Braet** [3]

[1] Intelligence in Processes, Advanced Catalysts & Solvents (iPRACS), Faculty of Applied Engineering, University of Antwerp, 2610 Antwerp, Belgium

[2] Energy and Materials in Infrastructure and Buildings (EMIB), Faculty of Applied Engineering, University of Antwerp, 2610 Antwerp, Belgium

[3] Department of Engineering Management, Faculty of Business & Economics, University of Antwerp, 2000 Antwerp, Belgium

[*] Correspondence: pieter.billen@uantwerpen.be; Tel.: +32-32658801

**Abstract:** To date, atmospheric concentrations of sulfur hexafluoride (SF$_6$) are the most potent among the greenhouse gases identified by the Intergovernmental Panel on Climate Change (IPCC) and are still rising. In the EU-28, SF$_6$ has been banned from several applications, however, an important exception is gas-insulated electrical switchgear (GIS) for which cost-effective and environmentally sound alternatives were unavailable when the F-Gas regulation was last revised in 2014. To date, after some recent innovations, we argue that the phasing out of SF$_6$ could spur the accelerated development of alternatives with a lower carbon footprint. In the EU-28, the SF$_6$ amount in switchgear is unclear. In this paper, we estimated the SF$_6$ amount to be between 10,800 and 24,700 t (with a mode at 12,700 t) in 2017, resulting in 68 to 140 t of annual emissions from operational leakage only, corresponding to 1.6 to 3.3 Mt of CO$_2$-eq. We additionally calculated the potential greenhouse gas savings over the lifecycle of one exemplary 145 kV gas-insulated switchgear bay upon replacing SF$_6$ by decafluoro-2-methylbutan-3-one (C5-FK) and heptafluoro-2-methylpropanenitrile (C4-FN) mixtures. Projecting these results over the EU-28, a phase-out scenario starting from 2020 onwards could reduce the carbon footprint by a median of 14 Mt of CO$_2$-eq, over a period of 50 years. Extrapolation to medium voltage could be assumed to be of a similar magnitude.

**Keywords:** carbon footprint assessment; electrical switchgear; sulfur hexafluoride; phase-out; transmission; distribution

## 1. Introduction

Sulfur hexafluoride (SF$_6$) is widely used in electrical switchgear equipment at medium and high voltage, due to its excellent insulation and arc quenching properties, such as its high stability, dielectric strength and heat dissipation, and its low boiling point [1,2]. Therefore, gas-insulated switchgears (GISs) using SF$_6$ can be made relatively small in size and can be used both for indoor and outdoor grid stations.

SF$_6$ is, however, a gas being cited to have a very high global warming potential (GWP) of approximately 23,500 kg of CO$_2$ equivalents per kg [3]. Nevertheless, next to electrical switchgear, SF$_6$ has formerly been used in a wide variety of other products, such as soundproof windows, and some of these applications are still marketed in some countries outside of the European Union. Within the latter, in 2006, a first EU F-Gas Regulation was adopted (Regulation (EC) No. 842/2006 [4]), with the aim of lowering potential climate change effects due to emissions of fluorinated gases. In its

original version, the use of fluorinated greenhouse gases was prohibited in various applications, such as domestic windows, footwear, tires, and others. This Regulation was renewed in 2014, introducing further restrictions on the use of fluorinated greenhouse gases (Regulation (EU) No. 517/2014 [5]). However, the renewed regulation states, "Equipment containing fluorinated greenhouse gases should be allowed to be placed on the market if the overall greenhouse gas emissions of that equipment, taking into account realistic leakage and recovery rates, are lower, during its lifecycle, than those that would result from equivalent equipment without fluorinated greenhouse gases." In our view, however, a complete life cycle assessment (LCA) including other impact categories should form the basis of alternatives, rather than greenhouse gases only. Additionally, it is specified that technological alternatives should be available, and not induce disproportionate costs. For these reasons, use of $SF_6$ in electrical equipment was still allowed, given the lack of realistically viable alternative at the time of the last legislative revision. Nonetheless, the use of $SF_6$ in GISs should be strictly monitored.

According to the National Inventory Submissions 2019 from the United Nations Framework Directive [6], the emissions of $SF_6$ in the EU-28, in 2017 only, was reported by the member states to be 286 t, of which 68 t or 2 Mt of $CO_2$-eq, was directly attributed to operational emissions from electrical equipment due to leakage. We stress that these are indirectly calculated reports, using various methods, probably leading to an underestimation. Twenty-four of the member states reported their estimated $SF_6$ stock in operational electrical equipment, as a base for calculating emissions, accounting for a total of about 10,400 t of $SF_6$ installed. Although monitoring and control measurements are required, few member states seem to have advanced nationwide accounting systems. Hence, this installed base constitutes a risk of over 240 Mt of $CO_2$-eq, if improperly released.

Since the previous revision of the F-Gas regulation (Regulation (EU) No. 517/2014), various new technological alternatives have been developed up to pilot scale and commercial demonstrations. Therefore, the continuous reconsiderations stated in the EU policy imply the need for thorough evaluation of the carbon footprint of technologies alternatives to $SF_6$-based GISs.

Currently, deployed commercial alternatives to $SF_6$-based GISs are in the medium voltage as well as in the high voltage range, up to 145 kV. For high voltage switchgear, the current main alternative to $SF_6$ is air insulated equipment, however, it has been demonstrated that its environmental performance is worse, given the higher areal footprint which prevents it from being installed in dense urban areas, and consequently, results in a suboptimal energy grid [7]. This equipment's size difference is in part due to the lower dielectric strength of the insulating medium provided in these systems.

Additionally, patent, professional and journal reports show accelerated efforts in the search for innovative and green switchgears. At medium voltage range fluid, solid as well as dry air insulated switchgears using vacuum circuit breakers are being developed [8–13]. Dry air insulated switchgears are also being developed for the high voltage range [14]. Two types of GIS technology have been developed for deployment both in the medium and in the high voltage range [8,15–19]. These are based on mixtures of $CO_2$ and $O_2$ with two compounds developed by 3M, namely decafluoro-2-methylbutan-3-one (C5-FK) and heptafluoro-2-methylpropanenitrile (C4-FN) (synthesis thereof described in [20,21]). The structures of C4-FN and C5-FK are given in Table 1. These mixtures are demonstrated to have both adequate insulation and arc quenching properties [1,22]. In addition to these switchgear mixtures, there is currently research ongoing on the application of 1,3,3,3-tetrafluoropropene (HFO-1234ze(E)) for both medium and high voltage switchgears [23,24]. A trifluoroiodomethane ($CF_3I$) mixture was researched as a possible alternative to $SF_6$ as well, yet it is unlikely to be implemented due its corrosiveness and suspected carcinogenicity [2,25].

Several individual manufacturers, such as General Electric [26], ABB [27], and Siemens [28], have recently announced the expansion of their respective $SF_6$-free technologies, also to higher voltage ranges, up to 420 kV.

**Table 1.** Chemical structures of the three investigated technological alternatives for gas-insulated switchgear and their typical mixture concentrations.

| Sulfur Hexafluoride (SF$_6$) | Decafluoro-2-Methylbutan-3-one C5-FK | Heptafluoro-2-Methylpropanenitrile C4-FN |
|---|---|---|
| 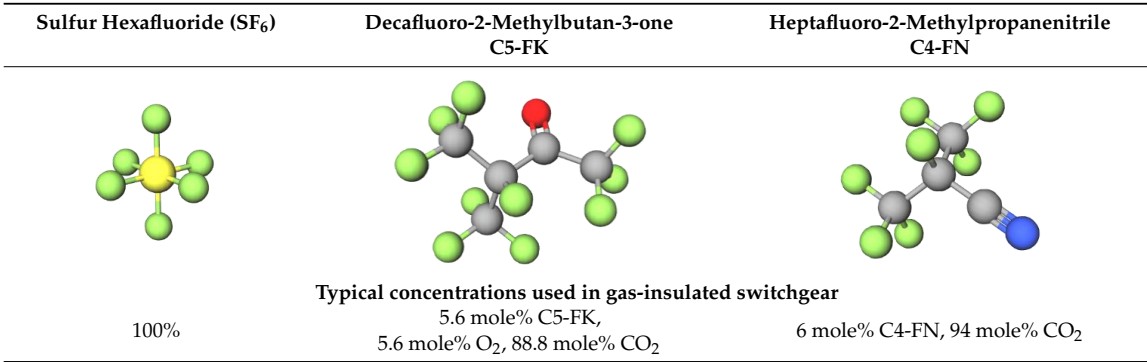 | | |
| | Typical concentrations used in gas-insulated switchgear | |
| 100% | 5.6 mole% C5-FK, 5.6 mole% O$_2$, 88.8 mole% CO$_2$ | 6 mole% C4-FN, 94 mole% CO$_2$ |

The goal of this paper is to project the lifecycle greenhouse gas savings associated with changing GIS systems from SF$_6$ to technological alternatives. The two major candidates for replacement of SF$_6$, next to dry air and air insulated switchgear which were evaluated in another study [7,13,29], are mixtures of C5-FK and C4-FN with CO$_2$ and O$_2$, as these are the closest to commercialization [26,27,30,31]. In this paper, typical mixtures of these two compounds with O$_2$ and CO$_2$ (Table 1) are benchmarked against SF$_6$ in exemplary high-voltage applications, based only on public records, reflecting the current state-of-knowledge. Next to the comparative ex-ante evaluation of greenhouse gas (GHG) intensity for a single 145 kV GIS bay, in this paper, we estimate the resulting effects of potential phase-out scenarios in terms of GHG savings for high-voltage applications on an EU-28 scale. Although calculations were only done for high voltage switchgear, given the superior availability of data, the results also gave insight in the extrapolation towards medium voltage applications. Potential differences in equipment size are anticipated, as have been shown in recent reports [32], however, these are not a subject in the present study due to the lack of available data for a detailed assessment.

## 2. Materials and Methods

### 2.1. Single High-Voltage Bay Comparative Lifecycle Carbon Footprint Assessment

In a first stage of this assessment, a single bay of SF$_6$-based switchgear was compared to a technologically similar bay using C5-FK or C4-FN mixtures, respectively. The functional unit was the operation of a single, new (i.e., state-of-the-art technology) 145 kV GIS bay, for 40 years. The amount of pure SF$_6$ in a traditional GIS was obtained from the publicly accessible g$^3$ calculator of General Electric Co. [33], which was 63 kg per GIS bay at an operating pressure 7 bar [34].

For a 145 kV GIS bay, the amount of C4-FN and CO$_2$ gas mixture is 30 kg [33] and its density is around half of the density of SF$_6$ [35]. We assumed a mixture of 6% mole/mole (equivalent to 22% mass/mass) of C4-FN in CO$_2$ at an operating pressure of 8 bar [34]. The pressure was increased to account for the lower dielectric strength of this specific mixture with respect to SF$_6$ [34]. This increase in pressure could be adjusted, which would require a higher percentage of C4-FN in the mixture. We accounted for such changes by a sensitivity assessment which investigated the effect of a changing molar ratio on the final results. Given the current state of technology, the overall mass ratio of alternative gases to SF$_6$ in the GIS was also subjected to sensitivity analysis.

A mixture of C5-FK/O$_2$/CO$_2$ on 5.6%/5.6%/88.8% molar basis (27%/3%/70% on mass basis) at 7 bar, has a dielectric strength of 77% of SF$_6$ [36] at 6 bar, and therefore the volume of mixture required for a 145 kV switchgear is about 30% higher than for a SF$_6$ switchgear. With this, the density of C5-FK at normal condition (10.7 kg/m$^3$ [37]) and assuming that all gases behave similar to ideal gases we estimated that for equivalent functionality, a switchgear with 33 kg of C5-FK/O$_2$/CO$_2$ mixture at 8 bar could replace a 63 kg SF$_6$-based switchgear at 7 bar.

In both cases, we did not include changes in the construction materials demand for the manufacturing of the GIS equipment itself in the assessment (in other words, we assumed the design of the equipment was similar in dimensions, and thus materials consumed). To date, data on the exact sizing of the equipment and impact on required building space are scarce, and the few data available are insufficient for a detailed assessment [32].

To build the emissions inventory, we distinguished the following four different phases: (1) gas mixture manufacturing, (2) filling of the switchgear bay, (3) operation while installing, and (4) decommissioning and recycling of the gas. Publicly accessible knowledge about the manufacturing of all gases is scarce; for $SF_6$ most studies cite Ecoinvent data [38], reporting a manufacturing impact of 122 kg $CO_2$-eq $kg^{-1}$. However, for this value primary sources date back to 1998, and therefore do not likely reflect state-of-the-art technology. For $CO_2$ and $O_2$, respectively, reported impacts for the manufacturing stage were 0.7 kg $CO_2$-eq $kg^{-1}$ and 0.6 kg $CO_2$-eq $kg^{-1}$ [38]. However, to our knowledge, no life cycle assessment of manufacturing and supply of C5-FK or C4-FN has been published in the literature thus far, as is the case for most fluorochemicals. As both gases were at the pilot scale production thus far, and no industrial analogue manufacturing impact was found, combined with the high uncertainty on $SF_6$ manufacturing impact, this stage was omitted from the study. Intrinsically, in doing this the assumed impact of manufacturing the gases or gas mixtures was equal. Assuming the carbon footprint of $SF_6$ of 122 kg $CO_2$-eq $kg^{-1}$ [38], and the low concentrations of C4-FN and C5-FK in their respective mixtures, this assumption was rather conservative. The low impact of manufacturing $CO_2$ and $O_2$ implied that the assumption used (i.e., omitted manufacturing stage), retained its conservative validity, up to five-fold, with respect to having high impact for the fluorochemicals as alternatives for $SF_6$. The resulting system boundary, including three different life cycle phases, is given in Figure 1, for the three alternative gas mixtures studied.

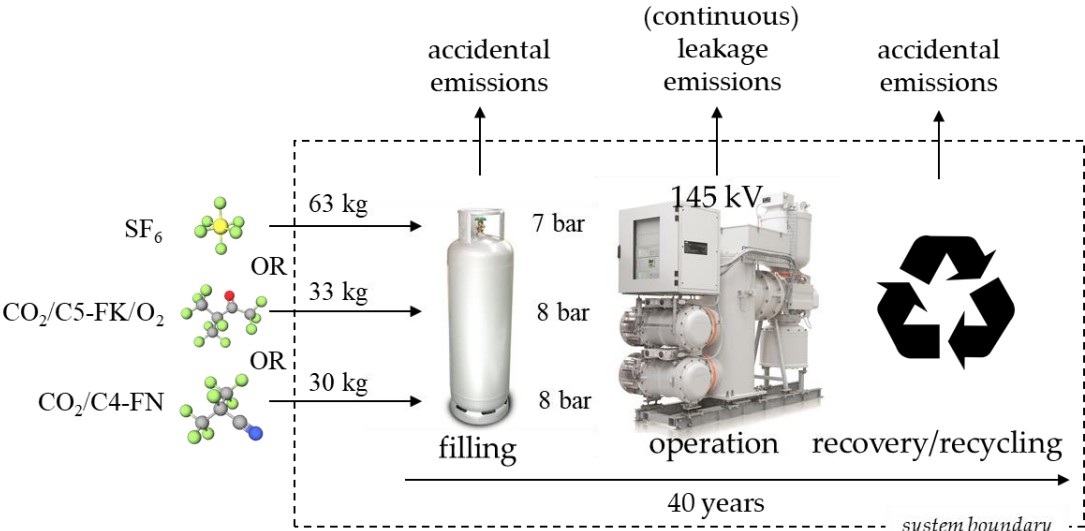

**Figure 1.** System boundary of the single 145 kV GIS bay carbon footprint comparison between the three investigated gas mixtures, demonstrated the three life cycle phases investigated, i.e., filling, operation and recovery/recycling. Filling and recovery/recycling are associated with instantaneous emissions, operational leakages are continuous over a 40-year period. This study only involved foreground analysis of the system, therefore, no background modeling was included, given the lack of data.

Emissions of the respective gases or gas mixtures during the initial filling and refiling of the GIS equipment (phase 2) were modeled using a modified PERT Probability Density Function (PDF) with a $\gamma$-factor (describing the weight to the mode) of one. The minimum emission factor was considered to be 0.1%, based on a value of 0.4% found during tests, whereby it was alleged about 15 years ago that a lower value would be possible with state-of-the-art equipment [39]. The mode or most likely

value was considered to be 0.5%, which is equivalent to the ABB environmental declaration [16] and the maximum was set at 1.6%, which is the maximum value targeted by some authors for 2020 [40]. However, the filling emissions were not standard operating procedure, as estimated leakages are always accidental. Therefore, they are inherently subject to high uncertainty.

Emissions occurring from the leakage during the operation of the GIS equipment (phase 3) were modeled using a modified PERT ($\gamma = 1$) PDF for the emission factors using parameters reported on recent equipment by switchgear manufacturers [16,17,39–44]. For $SF_6$, the minimum was set at 0.05% (as most manufacturers aim at a maximum of 0.1%), the mode was set at 0.1%, corresponding to the lower threshold for which $SF_6$ emissions should be actively monitored according to the revised F-Gas regulation [5]. The maximum of the PDF for the leakage emission factor was set at the industrial standard of 0.5% [45]. For the C4-FN/$CO_2$ and C5-FK/$O_2$/$CO_2$ switchgears that operated under higher pressure, these values were doubled, as a conservative estimate.

Given the regulatory advances of the last decade, research and development initiatives have been undertaken to increase the recovery and recycling of $SF_6$ from GISs [46,47]. We modeled the emission factor for phase 4, originating from the accidental emissions during extraction and transferring of the gas mixtures, with a modified PERT (mode weight equal to 1) distribution. Its mode was set at 2%, as stated by Glaubitz and Plöger [40] and as an often-reported value in UNFCCC reports for EU-28 countries [6]. The minimum value was assumed from the ABB environmental declaration [16] and the maximum was set at 5%, given that this was the maximum value reported by a country in the UNFCC reports [6]. Nevertheless, we argue that this value still seems rather low, given the vacuum that must be applied to the devices for such deep gas recovery. There is no apparent reason why the recovery and recycling of C5-FK or C4-FN mixtures would be any different to that of $SF_6$, hence the same PDF for the phase 4 emission factor was assumed.

The carbon footprint was assessed using a combination of deterministic data and probability density functions, as outlined above, by multiplying the direct emissions of the insulation gases or gas mixtures with their assumed global warming potential based on recent reports. The global warming potential of $SF_6$ as reported by the Intergovernmental Panel on Climate Change (IPCC) changed across their latest reports, with the latest value 23,500 kg $CO_2$-eq kg$^{-1}$ [3]. The global warming potential of C4-FN was less scrutinized, however, established at 2100 kg $CO_2$-eq kg$^{-1}$ by Owens et al. [48]. The global warming potential of C5-FK was shown to be lower than one [49,50]. The impact categories, other than global warming, were not taken into consideration in this work, given the absence of both fluoronitriles and fluoroketones in impact assessment methods. Nonetheless, although exposure risks were assessed according to the REACH framework, future work should demonstrate potential toxicity effects in a life cycle perspective.

## 2.2. Estimation of EU-28-Wide Installed Amount of SF_6 Gas in Gas-Insulated Switchgear Equipment

Emissions of $SF_6$ were estimated from UNFCCC [6], which is an international environmental treaty to report on GHG emissions via a bottom up approach based on IPCC guidelines. In analogy to our single bay assessment, the UNFCCC method distinguishes production, use, and decommissioning. Data on the amount of $SF_6$ in produced, operating, and decommissioned switchgears is collected as a first step of the bottom up calculations [51]. Consequently, although not strictly required, many countries do explicitly or at least implicitly mention their estimated amounts of $SF_6$ in electrical equipment [52]. The UNFCCC reports emission data as far back as 1990 and until two years prior to the last publication (hence the 2019 report has data on 2017 emissions). The total reported stock for the EU-28, excluding four countries, in 2017, is 10,428.5 t $SF_6$, however, extrapolation to the entire EU-28 using the installed net electricity generating capacity [53] leads to an estimate of 10,800 t, for 2017.

Nevertheless, the comparison of UNFCCC reported data, which are predominantly a result of indirect accounting methods, with the results of the more generic modeling method, the Emissions Database for Global Atmospheric Research (EDGAR) shows a large discrepancy. The EDGAR modeled, from 2000 until 2010, the $SF_6$ atmospheric emissions (up to version v4.2 FT2010 [54]) based

on international annual technological statistics combined with geographical information systems modeling, in a more generic and transversal way than UNFCCC [55]. Similar to earlier accounts [55,56], we appreciate, in Figure S3 of the Supplementary Materials, that the EDGAR calculations are much more in line with empirically derived data. This is demonstrated by benchmarking the estimated global and regional emissions to top-down inverse modeling of regional emissions based on atmospheric measurements of the AGAGE project [57–61]. Moreover, multiple reports, therefore, have indicated that UNFCCC emissions data are likely underreported [62–65].

To date, however, the public data are inconclusive, whether the emission factors used in UNFCCC reporting methods are underestimated, or the underlying assumptions about the installed stock of $SF_6$. Therefore, we approached the installed stock of $SF_6$ in electrical equipment as a probability density function (PDF) with the minimum and maximum defined by two scenarios; either the installed stock of UNFCCC is correct at 10,800 t (assuming the emission factors are strongly underestimated), or the average emission factor of UNFCCC reporting (0.57%, more details are found in the Supplementary Materials Section S1) is correct, leading to an installed stock of 24,700 t, as EDGAR emission estimates were about twice as high as UNFCCC estimates for the last reported years (2007 to 2010). In our view, the most likely stock value (mode of the PDF) is 12,700 t (equivalent to almost 300 million t of $CO_2$-eq, if ever released), which is an extrapolation of the German stock figure on the basis of net electricity generating capacity [66] (see Section S3 of the SI). The German reporting, since 2011, has been based on detailed sales data of manufacturers, including imports and exports, constituting a reliable mass balance [6], and its extrapolation seems to have used the most likely value for the EU-28.

The resulting probability density function of installed $SF_6$ stock in electrical equipment was aggregated; the data contains both medium voltage and high voltage equipment. Because we firstly investigated an exemplary high voltage switchgear in this study, represented by the 145 kV GIS bay functional unit, an apt estimator for the EU-28 wide HV share was required. To this end, we used transmission and distribution grid operator data [67–80] next to direct UNFCCC reports for the five EU countries with the highest $SF_6$ stock (accounting for about 80% of the total EU-28 stock), and determined an average share of high voltage $SF_6$ stock of 45% of the total. This estimate was also subject to a skewed probability distribution as a result of uncertainty on the distribution grid data quality (see the discussion in the Supplementary Materials, Section S4).

*2.3. Consequential Time-Series Modeling of Greenhouse Gas Savings in Phase-Out Scenarios*

A projection of the EU-28 stock of $SF_6$ gas in HV switchgear was made by regression of the stock changes with the installed net electricity generating capacity (NEGC) for the period 2000 to 2017. Earlier, a linear relation was found between the amount of $SF_6$ installed and the grid capacity in China [81]. In the present study, the regression (stock $SF_6$/NEGC) was done using data of Eurostat [53] for the grid capacity, and the stock of $SF_6$ directly taken from UNFCCC [6] reports. The resulting regression was applied to the PDFs of the stock; as only linear transformations were applied to the primary data. The expansion of the grid, and hence the growth of $SF_6$ stock in GISs under unchanged policy, was modeled by applying the $SF_6$/NEGC regression function to the projection of the electricity installed capacity of the energy reference scenario of the European Commission of 2016 [82]. The latter seems conservative, as no expansion is expected in this scenario for the next decade.

The projected growth of GIS stock is over an analysis period of 50 years (2020 to 2070), whereby we developed scenarios for a phase-out of $SF_6$ initiating in 2020. A phase-out scenario means that all newly installed switchgears should be based on an $SF_6$-free alternative technology, whereas existing installations are continued to be used and maintained until end-of-life. The lifetime expectancy of each individual HV GIS bay is assumed to be 40 years. We did not assume a refilling of the bays to account for the gas lost by leaking which, again, was conservative toward the lower GWP gases. New GIS installations are installed either when current $SF_6$-based GIS bays are at end-of-life and need replacement, or when the electric grid capacity is expanding. The EU reference scenario has projected slight decreases of net electricity generating capacity for some periods [82], however upon an expected

decrease of the net electricity generating capacity, no decrease of the switchgear capacity was assumed. This was very conservative in our view, given the further developments in the grid, as well as the potential for technology expansions. A detailed quantitative description of the calculation methods is given in Sections S6 and S7 of the Supplementary Materials.

## 2.4. Uncertainty Analysis

Where possible throughout the study, we applied stochastic methods on data, as outlined in the sections above. After performing a sensitivity analysis in which we studied the effects of varying each variable independently in a +10% and –10%, we could detect the ones that had a greater influence in the final result. In the case that there was sufficient information or informed criteria, these variables were modeled as probabilistic. In other cases, a parametric sensitivity analysis was performed.

The propagation of uncertainties (and corresponding PDFs) were carried out using Monte Carlo calculations, taking 30,000 samples with the Latin median hypercube method. This technique spreads the sample points more evenly across all possible values than Monte Carlo sampling, by selecting the median values of 30,000 intervals of equal probability instead of generating a random sample of 10,000 points [83]. Calculations were performed using Analytica software. Table 2 gives a summary of the PDFs for the variables used as input to the analysis. Variables that did not figure in the table are considered as deterministic values in the analysis, as for them a reasonable PDF could not be argued. For some of these variables that have an important impact on the result, a parametric sensitivity analysis was performed.

**Table 2.** Description of probabilistic parameters used with respect to life cycle inventory data.

| Parameter | Uncertainty Background | PDF Type | Description |
|---|---|---|---|
| EF filling [16,39,40] | Variety across sources | Modified PERT ($\gamma = 1$) | SF$_6$: 0.1%, 0.5%, 1.6%<br>Alt [a]: 0.1%, 0.5%, 1.6% |
| EF operation (leaks) [5,16,41–44] | Variety across sources | Modified PERT ($\gamma = 1$) | SF$_6$: 0.05%, 0.1%, 0.5%<br>Alt [a]: 0.1%, 0.2 %, 1% |
| EF recovery/recycling [6,16,40] | Variety across sources | Modified PERT ($\gamma = 1$) | SF$_6$: 1%, 2%, 5%<br>Alt [a]: 1%, 2%, 5% |
| EU-28 total SF$_6$ stock in GISs | UNFCCC vs. EDGAR data | PERT negatively skewed | Min = 10800, mode = 12,700, max = 24,700 |
| Share of HV SF$_6$ stock | MV stock uncertainty | Modified PERT ($\gamma = 1$) | 38.5%, 45%, 87.3% |
| Intercept regression SF$_6$/IC [b] | Regression | Normal | $\mu = -6930$ $\sigma = 7$ % |
| Parameter coefficient regression SF$_6$/IC [b] | Regression | Normal | $\mu = 0.017$ $\sigma = 3.2$ % |

[a] Alt, alternative technology, including either C4-FN or C5-FK mixtures and [b] IC, grid installed capacity, equal to the maximum of historical NEGC.

## 3. Results

### 3.1. Single 145 kV GIS Bay Comparative Carbon Footprint Assessment

Replacement of the insulation gas of a single 145 kV GIS bay with a lifespan of 40 years, seems to result in a very high potential of emissions savings during the use phase, i.e., as a result of operational leakage, given the lower GWP of the alternative technology compounds (Figure 2). Additionally, since similar dielectric strengths can be accomplished using mixtures of C4-FN and C5-FK with $CO_2$ (and $O_2$) at increased pressures, the total amount of fluorinated gases to be produced is significantly lower than the amount of SF$_6$ for comparable performance. Figure 2 shows the carbon footprint of the three investigated lifecycle phases (filling, operation, and decommissioning) of different GIS systems, with gas composition and amounts, as well as emission factors, as outlined above. The projected savings by using C4-FN or C5-FK technology from the direct gas emissions during filling, use, and

recovery/recycling are completely due to the lower GWP of the respective gas mixtures. The impact of the three calculated lifecycle phases for alternative technologies is negligible as compared with the $SF_6$-based GISs. Although no public peer reviewed manufacturing (cradle-to-gate) LCA results for the production phase of most fluorochemicals are available, and therefore the impact of this phase is highly uncertain and not accounted for, this means that the manufacturing impact of the total gas mixture should be about 170 t of $CO_2$-eq higher than that of $SF_6$ to exceed the overall carbon footprint of a current individual GIS bay. Moreover, if we assume a current manufacturing carbon footprint of the 63 kg of $SF_6$ in one GIS bay, the resulting impact of 8 t of $CO_2$-eq (122 kg $CO_2$-eq per kg, Ecoinvent) is very low with respect to the overall lifecycle carbon footprint. Therefore, it is very unlikely that the manufacturing impact of the alternative technology gas mixtures, containing less than 25% m/m of the fluorochemicals, has a manufacturing impact of over 20 times that of $SF_6$. Therefore, it is safe to conclude that the manufacturing stage has no influence on the general trend that $SF_6$-free technology has the potential to reduce the carbon footprint of HV GIS bays. This is in agreement with Rabie and Franck [55] who stated that the impact of manufacturing gas becomes irrelevant in comparisons between fluorochemicals involving $SF_6$ that have such high reported GWP.

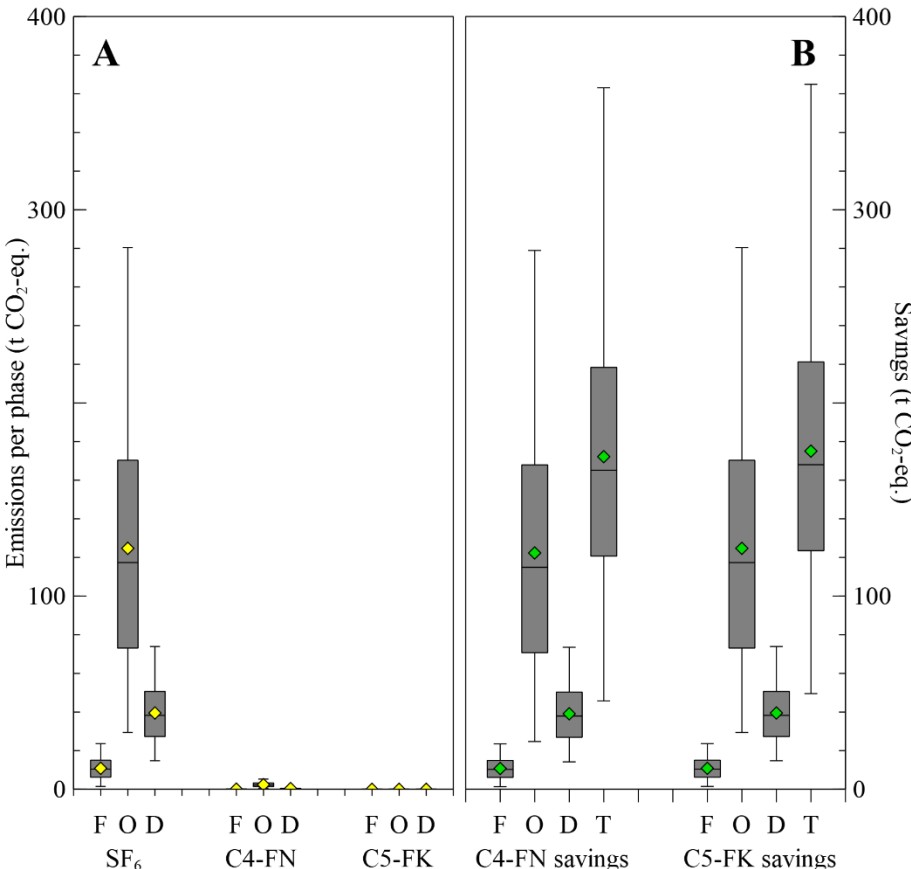

**Figure 2.** (**A**) Comparison of the carbon footprint, in metric tons of $CO_2$-eq, per functional unit of a single 145 kV GIS bay over a lifetime of 40 years of operation, for $SF_6$-based, C4-FN-based, and C5-FK-based technology; (**B**) Savings for both alternative technologies with respect to $SF_6$ systems are also shown. Results are disaggregated into filling (F), operation (O), and decommissioning (D), or shown as total savings (T). The boxplots are composed of the median, interquartile range, and minima and maxima, with the diamonds representing the averages. A similar graph with logarithmic scale is shown as Figure S5 in the Supplementary Materials.

### 3.2. Cumulative Greenhouse Gas Savings in the EU-28 (HV) Transmission Grid in SF$_6$ Phase-Out Scenarios

The total SF$_6$ stock in GISs in the EU-28 was projected using the established regression between the installed net generating electricity capacity in the period from 2000 to 2017 on an annual basis, and the UNFCCC reported stock of SF$_6$ in electrical equipment (Figure 3A). Using this regression, we projected the PDF of the EU-28 SF$_6$ stock of 2017, toward 2070, using projections of the net electricity generating capacity of the EU [82], as shown in Figure 3B. This projection concerns both the distribution (MV) and transmission (HV) stock of SF$_6$. The total stock of SF$_6$ is expected to stagnate between 2020 and 2035, as a result of a projected stagnation of the net electricity generating capacity in this period. Therefore, no expansion of the amount of installed GIS bays is expected. This is rather conservative, as currently grids are becoming more decentralized, electricity consumption is rising, and industrial electrification is intensifying. Nonetheless, bays at the end of their 40-year lifespan are replaced by alternative technology in a phase-out scenario. The annual amount of SF$_6$ replaced in this manner is nevertheless considerable, given the sharp historical SF$_6$ stock increase in the period from 1980 to 2000 (Figure 3B). The uncertainty, predominantly a consequence of the aforementioned limited reliability of UNFCCC [6] reports, is represented by the interquartile range (IQR) and the 95% central range (95CR).

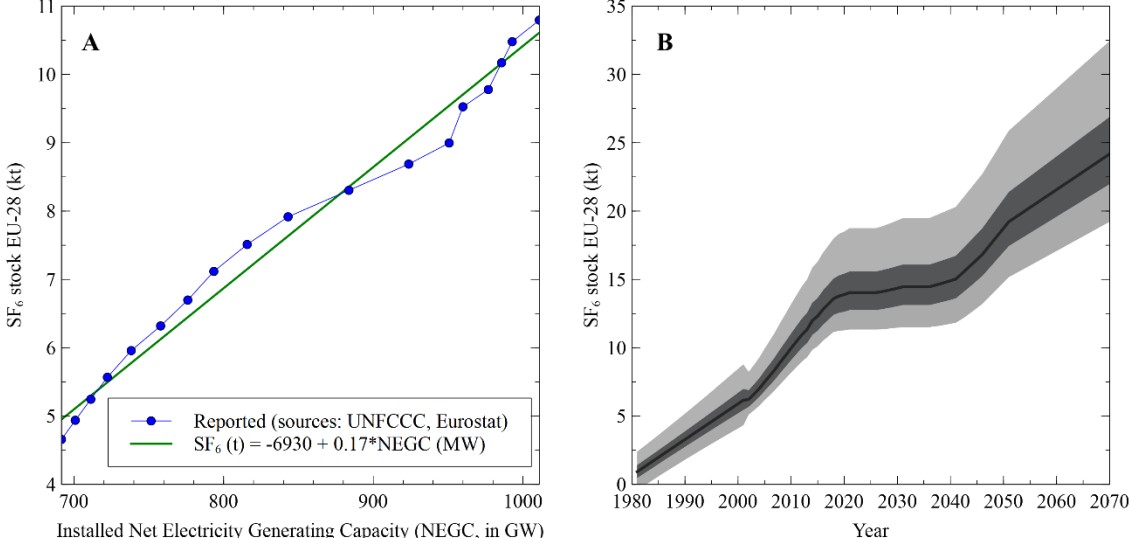

**Figure 3.** (**A**) Regression of the increase of SF$_6$ stock at the EU-28 scale resulting from an expansion of the net electricity generating capacity of the grid, based on UNFCCC (2019) and Eurostat (2019) data, with implications for the parameters of the stock probability distribution; (**B**) Projection of this regression on the outlook for the electricity grid until 2070. The lines represent the median, the dark and light shades in (**B**) represent the IQR and 95CR, respectively.

The projections, from Figure 3B, allowed us to subsequently estimate the replacement of SF$_6$ in the GISs in a phase-out scenario, starting from 2020, for the transmission (HV) grid, constituting about 45% of the total SF$_6$ stock. The example of the 145 kV single-bay assessment is taken as representative for the HV grid. Assuming at end-of-life (i.e., after 40 years) all SF$_6$-based GISs must be replaced by alternative technology (either C4-FN or C5-FK) and assuming that expansions of the grid, upon an increase of the net electricity generating capacity (Figure 3B), should also be done using SF$_6$-free technology, the potential consequences of such a phase-out scenario become apparent, as shown in Figure 4. Herein, we show the identified GHG emissions from either the SF$_6$-based GIS technology scenario (i.e., business-as-usual, continuation of state-of-the-art SF$_6$-based GISs) versus those scenarios in which alternative technologies are introduced, using the single 145 kV GIS bay comparisons, shown in Figure 2. The large year-to-year variations of the emissions are due to the various timings in which SF$_6$ technology is either introduced (emissions of filling) or taken out of the grid (emissions from recovery/recycling), see the online Supplementary materials (Figure S6) for more background. Upon

cumulating the differences of the annual emissions, it is clear that the savings could amount to about 14 Mt of $CO_2$-eq, as the median of a PDF between 4 and 31 Mt of $CO_2$-eq as the 95% central range. The potential savings achieved by both $SF_6$-free technologies are nearly overlapping. Postponing the phase-out by five years, starting in 2025, would decrease the cumulated savings in 2070 to a median of 12.5 Mt of $CO_2$-eq (Figures S7 and S8 in Supplementary Materials). Such a scenario is more likely, allowing more time for further technological development and deployment.

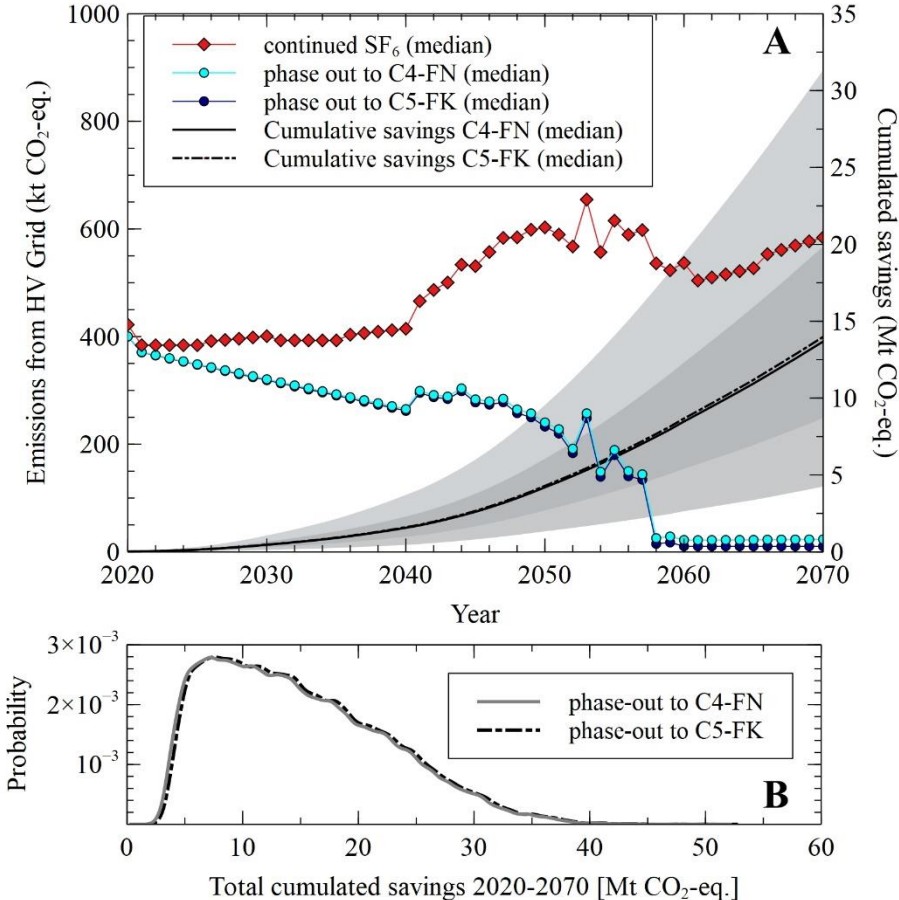

**Figure 4.** Annual aggregated emissions (not actual emissions, but assuming the grid is state-of-the-art technology) from the EU-28 transmission grid (HV GIS bays), in kt of $CO_2$-eq in three different scenarios; business as usual, phase-out starting 2020 toward C4-FN technology, and phase-out starting in 2020 toward C5-FK technology (left axis of A), and cumulated savings of both alternative technologies with respect to continued $SF_6$ technology in Mt of $CO_2$-eq (right axis of A). For clarity, in (**A**) probability distributions are not given for the annual emissions values, but shown in the cumulative savings as IQR (dark shaded) and 95CR (light grey shaded), only for C4-FN scenario as both phase-out scenario savings nearly overlap, as appearing from their final PDFs (**B**). The latter represent the modeled probabilities only for total cumulated carbon footprint.

## 4. Discussion

As shown in the previous section (e.g., Figure 3B), there is a high uncertainty related to the amount of $SF_6$ installed, given the past discrepancies between UNFCCC reports and backward modeling from atmospheric concentrations. Additionally, our analysis strongly relies on projections of the electricity grid capacity [82], which is by definition uncertain, as well as on the assumption that the past relation between net electricity generating capacity and installed $SF_6$ stock can safely be extrapolated. The first of these intrinsic uncertainties is captured in our modeling, whereas the latter two are not, given our limitations to deal with potential market or technological disruptions. Nevertheless, results for the

single 145 kV GIS bay, representative for a transmission grid GIS, are independent of these uncertainties, and therefore are considered to be as robust. The savings per individual $SF_6$-based GIS bay replaced by C4-FN or C5-FN technology range from about 50 to 360 t of $CO_2$-eq, with median around 170 t of $CO_2$-eq. Given the technological uncertainties related to novel technology, in our modeling we doubled the emission factors for operational leakage, adding to the robustness of our findings. The total savings could amount to 14 Mt of $CO_2$-eq (median) over a period of 50 years in a phase-out scenario towards either C4-FN or C5-FK as alternative technologies. In spite of potential savings being rather small when related to the total impact of electricity at point of consumption, the technology is available for rapid market penetration. However, we do stress that other impact categories were not taken into account and should be further studied.

Recent industrial reports have indicated that the emission factors for state-of-the-art HV GIS technology, relevant for the future projections made, are currently approximating those of MV GIS technology. However, we did not model the replacement ratio on a molar or mass base. However, although not explicitly modeled in this work, if replacement ratios are not too different from those of HV GIS bays, the similarity of emission factors implies that similar GHG savings could be expected. More uncertain is the potential replacement scenario for the distribution grid, however, it is assumed, given its share of about 55% of all currently installed $SF_6$, that the potential cumulated savings on an EU-28 scale are of the same order as those for the transmission grid. Additionally, the current trend toward decentralized electricity generation, causes accelerated expansion of the distribution grid, and hence medium voltage switchgear.

## 5. Conclusions

Alternatives to $SF_6$ for gas-insulated electrical switchgear, such as those based on fluorochemicals such as decafluoro-2-methylbutan-3-one (C5-FK) and heptafluoro-2-methylpropanenitrile (C4-FN), are considered to be emerging technologies, however, the significant difference in reported global warming potential of these gases and their mixtures clearly results in a large potential reduction of the overall carbon footprint. An exemplary and conservative analysis on a single 145 kV GIS bay shows savings between 50 and 360 t of $CO_2$-eq over its lifespan, when switching to C5-FK or C4-FN instead of state-of-the-art $SF_6$ technology. Although studied under large uncertainty, given the early technological stage, the results are robust toward either higher emission factors, as well as higher manufacturing impact of the C5-FK and C4-FN technologies. However, the latter should be corroborated by additional research, as should these findings be expanded to other impact categories.

The single bay assessment was projected to the entire transmission grid in the EU-28, using the estimated total amount of $SF_6$ in electrical equipment, and a growth thereof calculated through a correlation with grid capacity. A potential phase-out scenario for high voltage equipment as from 2020, assuming end-of-life equipment replacements and grid expansions should be done by C5-FK or C4-FN technologies, results in a total savings of 4 to 31 Mt of $CO_2$-eq, with the median at 14 Mt of $CO_2$-eq, over a subsequent period of 50 years. A postponed phase-out to 2025, taking into account the current technological readiness, would still result in a median of 12.5 Mt of $CO_2$-eq saved over a period until 2070. The broad distribution is caused predominantly by the uncertainty of the installed stock, next to the uncertainties of the emission factors. In spite of not having explicitly modeled medium voltage equipment, savings of the same order are assumed for the distribution grid gas-insulated switchgear.

**Supplementary Materials:** The following are available online at http://www.mdpi.com/1996-1073/13/7/1807/s1, supplementary information on calculation methods, data sourcing and analogous results, including Figures S1–S8 and Tables S1–S10.

**Author Contributions:** Conceptualization, P.B.; methodology, P.B. and J.B.; formal analysis, B.M. and M.L.; investigation, B.M. and M.L.; resources, P.B.; writing—original draft preparation, P.B., B.M., and M.L.; writing—review and editing, J.B.; visualization, M.L.; supervision, P.B. and J.B. All authors have read and agreed to the published version of the manuscript.

**Funding:** This study was funded by 3M Belgium BVBA.

**Acknowledgments:** Baptiste Audenaert is acknowledged for his contributions in data gathering and processing of SF$_6$ emissions and stock.

**Conflicts of Interest:** The funders had no role in the design of the study; in the collection, analyses, or interpretation of data; in the writing of the manuscript, or in the decision to publish the results. Any views and opinion expressed in this paper are those of the authors, and do not necessarily reflect those of 3M. The presented research does not support any statement in favor or against the impact of anthropogenic GHG emissions on climate change.

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
