# Peer review of "Replacing SF6 in Electrical Gas-Insulated Switchgear: Technological Alternatives and Potential Life Cycle Greenhouse Gas Savings in an EU-28 Perspective"

_energies, doi:10.3390/en13071807_

Round 1
Reviewer 1 Report
My suggestions are as below;
The authors must include flow chart or a figure to depict system boundaries of the LCA carried out.
LCA has been carried out without following ISO standards, why? it is essential for validation.
Inventories of LCA input and output must be explained and added into the study.
Results and discussion must be improved, mere three figures are less for LCIA. A detailed analysis is required to define the novelty of the study.
Author Response
General response to the reviewers: In addition to our detailed addresses to the reviewer comments below, we also have checked the manuscript for remaining linguistic or typographic issues. Furthermore, we like to indicate that the focal point of this paper is the projection of potential carbon footprint savings toward a complete EU grid system, rather than on a full LCA. Moreover, the state of the technology is very young, therefore not enough data are present for a full LCA. This has been appreciated by reviewer 2 and 3, who only have minor comments, if any. We thank both the editor and the reviewers for their consideration.
My suggestions are as below;
The authors must include flow chart or a figure to depict system boundaries of the LCA carried out.
Response: On line 152, we have added figure 1, which describes the system boundaries in a clear way.
LCA has been carried out without following ISO standards, why? it is essential for validation.
Response: All crucial steps of the ISO standard 14040, being goal and scope definition, life cycle inventory, impact assessment and interpretation, were included in the study. The goal is defined on line 95, the functional unit on line 111 and system boundaries now also included in Figure 1, line 152. The system boundary deviates from the ISO standard, because the analysis is gate-to-gate, not cradle-to-grave, as the standard demands. However, the rationale for this, because of the lack of data at this early point in the development, is clearly indicated in the text, line 134-151. The inventory is established in section 2.1 and summarized in Table 2 (we extended the caption of this table for clarity with “with respect to life cycle inventory data”). Furthermore, the analysis was restricted to greenhouse gases, since other data are uncertain at this point and the study is not toxicological in nature. This is indicated in the text. With respect to the potential impact on climate change, results are clearly presented and interpreted with regard to the study’s goal, in the discussion section.
We do want to stress that also a key aspect of this study is not merely the LCA of a single bay GIS, but rather the potential effect of changed energy policy on greenhouse gas emissions in a large region such as the European Union.
Inventories of LCA input and output must be explained and added into the study.
Response: All data required to reproduce the results of this study are now presented in the combination of Table 1, Figure 1, and Table 2. Moreover, in the supporting information section S6 and S7, the entire calculation method leading to the final results is reported (this is now also clarified in line 264-265: “A detailed, quantitative description of the calculation methods is given in the sections S6 and S7 of the SI.”).
Results and discussion must be improved, mere three figures are less for LCIA. A detailed analysis is required to define the novelty of the study.
Response: We would like to point out that the figures as presented fully allow to support the findings related to the objective of our research. The main objective is to support future policy decisions related to the carbon footprint of SF6 alternatives, therefore the results comprise various figures on carbon footprint analysis outcomes. Furthermore we stress that we did every effort to consolidate as many as possible our results into a limited set of figures that contain a lot of information, in order to present a concise paper. Figures that are less relevant to understand the main point of the paper are shown in the supporting information.
The novelty of the study is that in no earlier publication, a detailed assessment of the amount of SF6 in electrical switchgear was given, nor an assessment of the potential replacements thereof on a regional/continental scale.
Reviewer 2 Report
As alternatives for SF6 in in gas insulated electrical switchgear, the authors considered C5-FK and C4-FN. The most significant result is that the total amount of fluorinated gases to be produced for C5-FK and C4-FN are significantly lower than the amount of SF6 for comparable performance. Conservative analysis on a single 145 kV bay showed that savings between 50 and 360 t of CO2-eq. when switching to C5-FK or C4-FN instead of state-of-the-art SF6 technology.
The results are quite interesting and the related discussions are well organized with supplementary materials. Therefore, I think it can be accepted.
Author Response
As alternatives for SF6 in in gas insulated electrical switchgear, the authors considered C5-FK and C4-FN. The most significant result is that the total amount of fluorinated gases to be produced for C5-FK and C4-FN are significantly lower than the amount of SF6 for comparable performance. Conservative analysis on a single 145 kV bay showed that savings between 50 and 360 t of CO2-eq. when switching to C5-FK or C4-FN instead of state-of-the-art SF6 technology.
The results are quite interesting and the related discussions are well organized with supplementary materials. Therefore, I think it can be accepted.
Response: We thank the reviewer for the feedback given, and did not see reasons for changing the manuscript based on his/her comments.
Reviewer 3 Report
The article is very clear and well organized. I have only some doubt about the consistency with the journal aims.
Figure 1 should be improved because the scale doesn’t allow to appreciate the differences. Maybe you can use a logarithmic scale or a different scale for C4-FN and C5-FK. Also, the differences in savings are not appreciable.
The authors focused only on the GHG emissions, for climate change purposes. Have they evaluated other environmental aspects (ecotoxicity, human health)?
How the authors addressed the end of life of the gases?
Author Response
Figure 1 should be improved because the scale doesn’t allow to appreciate the differences. Maybe you can use a logarithmic scale or a different scale for C4-FN and C5-FK. Also, the differences in savings are not appreciable.
Response: We deliberately chose to use the exact same axis scale for Figure A and Figure B, to demonstrate that almost the complete impact of SF6 could be saved by switching to alternative gas mixtures. Additionally, by switching to a logarithmic scale, we saw that (1) box plots become somewhat distorted, and more importantly (2) absolute readouts of the savings become far more difficult. We acknowledge that by doing this (using the linear scale), comparative information between the fluoroketone and the fluoronitrile is not apparent. Nevertheless, such comparison is not the object of this study, which is rather investigation of the effects of a potential phase-out of SF6.
However, we accounted for this remark by including also a similar figure, completely using a logarithmic scale, to the supplementary information (Figure S5), and referring to it in the main text figure’s caption (line 319-320).
The authors focused only on the GHG emissions, for climate change purposes. Have they evaluated other environmental aspects (ecotoxicity, human health)?
Response: Other environmental aspects were not evaluated, given the lack of data at this point to translate early-stage toxicological tests* to impact assessment methods (and hence characterization factors for e.g. 1,4-dichlorobenzene equivalents). As the authors are no toxicologists, we are not qualified to make such judgements, and we now explicitly stated why we did not include other aspects in line 194-197: “Impact categories other than global warming were not taken into consideration in this work, given the absence of both fluoronitriles and fluoroketones in impact assessment methods. Nonetheless, although exposure risks are being assessed according to the REACH framework, future work should demonstrate potential toxicity effects in a life cycle perspective.”
*see e.g. Christophe Preve, Romain Maladen, Gérard Lahaye, Thibauld Penelon, Myriam Richaud, et al. Hazard study of MV switchgear with SF6 alternative gas in electrical room. 24. International Conference and Exhibition on Electricity Distribution (CIRED 2017), Jun 2017, Glasgow, United Kingdom. pp.198-201, ff10.1049/oap-cired.2017.0385
How the authors addressed the end of life of the gases?
Response: This is clarified in lines 177-187, where we base the analysis on recovery and recycling reports from conferences as well as industrial open declarations, and the UNFCCC reporting. This leads to a probability distribution on the amount of accidental emissions during this operation of extracting and transferring the gas (mixtures). We added the following in line 179: “originating from accidental emissions during extraction and transferring of the gas mixtures”.
Round 2
Reviewer 1 Report
The authors have modified paper well. It can be published, if editor finds fit for publication.